# [RE] A Reproducibility Study on Scene-Graph Generation from 3D Point Clouds: Hybrid Approach with Clip, 2D Image Semantics, and 3D Geometry

## Abstract

## Reproducibility Summary

### Scope of Reproducibility

This paper scrutinizes the reproducibility of VL-SAT and multimodal learning systems for 3D semantic scene graph prediction. Leveraging visual (ViT, CLIP) and linguistic semantics, our study replicates top-k accuracy results and explores models like SGFN, and SGGPoint. We assess the impact of the CLIP adapter, 2D image semantics, and conduct hyperparameter tuning. Additionally, the ablation study investigates node and edge collaboration, and the influence of a multi-head self-attention network within the VL-SAT architecture, enhancing understanding of these critical components.

### Methodology

We use the open-source code released by the authors to generate datasets, create point cloud data, and train and validate samples for VL-SAT. Our implementation covers 150 3D reconstructed indoor scenes from the original 1553, maintaining the 160 object classes and 26 predicate types as outlined in the paper. Additionally, we collaborate with the authors to integrate code for models SGFN, and SGGPoint into our existing code-base. Expanding upon the methodology, we meticulously implement the provided specifications, addressing any gaps to ensure a comprehensive pipeline supporting all experiments. Our experimentation uses computational resources provided by an NVIDIA GeForce GTX 3090 GPU, totalling 100 GPU hours for training. Moreover, we secure access to GPU compute resources through collaboration with the ML Collective team.

### Results

Upon executing the authors' provided code, we encountered the necessity for substantial modifications and additions, including the incorporation of numerous files. Following these adjustments and the addition of essential segments, we conducted reproducibility tests, ablation studies, and hyperparameter tuning. Consequently, our results largely support the main claims of the paper within a significant subset of experiments. However, there are notable discrepancies in many of the actual values obtained compared to those reported. Hence, we conclude that while the paper's findings are largely replicable, achieving precise reproducibility of results requires additional efforts due to the extensive changes and additions required in the provided code.

### What was easy

We found it easy to discern the primary assertions of the paper and the corresponding experimental evidence. Furthermore, the availability of the authors' open-source implementation facilitated ease in training the model, conducting ablation studies, and fine-tuning hyperparameters.

**What was difficult**

Configuring the datasets presented challenges primarily due to the absence of pinned dependencies, and the lack of code for generating 3D datasets resulted in delays in conducting experiments. Additionally, identifying the sources of discrepancies in our findings proved challenging, compounded by the inaccessibility of training curves and model weights or checkpoints. These limitations hindered our ability to precisely replicate the reported results and necessitated additional efforts in troubleshooting and refining our implementation.

**Communication with original authors**

At the initiation of our research endeavour, we diligently maintained ongoing communication with the authors through email channels which benefited us with their valuable insights and resources, thereby enhancing the depth and scope of our study. However, subsequent to the integration of code for the models under investigation, our attempts to engage in further correspondence with the authors were met with silence.

# 1 Introduction

Scene understanding ( 1, 2, 3, 9, 5, 6, 9, 8, 10, 11, 8) plays a pivotal role in various fields such as computer vision, robotics, and augmented reality (AR). It involves the comprehension and interpretation of complex visual scenes, enabling machines to extract meaningful information from images or videos. This capability is crucial for tasks such as object recognition, scene segmentation, and activity recognition. One fundamental concept in scene understanding is the scene graph, a data structure representing objects in a scene and their relationships. It provides a structured representation of visual scenes, capturing not only the objects present but also the spatial relationships between them. This hierarchical representation allows for richer semantic understanding and facilitates high-level reasoning about the scene. Initially, scene graph generation from images involved detecting and classifying objects in the scene and then detecting and classifying relationships given an image and predicted objects. But now, the most common and effective way for scene graph generation is detecting objects and then evaluating edge and node features to infer relationships between them. All the models, such as MMGNet, SGFN, and SGGPoint, employed in Wang et al. 1, adopt the latter approach for scene graph generation. The process can be succinctly described as follows: an input image undergoes processing through a Faster R-CNN to detect objects, which then form a complete graph. Subsequent message passing and classification layers (for node and edge classes) yield the scene graph $P(G|I)$, where $G$ represents the graph and $I$ the input image.

Formally, the generation of a scene graph is given by:

$$P(G|I) = P(O, R|I) = P(O|I) \cdot P(R|I, O) \tag{1}$$

This equation denotes the initial detection and classification of objects $O$ within a point cloud, followed by the detection and classification of relationships $R$, given the point cloud and predicted objects.

Further refining, the process involves:

$$P(G|I) = P(V, E|I) \cdot P(O, R|V, E, I) \tag{2}$$

Here, objects are first detected, then object features $V$ and edge features $E$ are classified, leading to the classification of objects and their edges.

Predicting 3D semantic scene graphs in point clouds is challenging due to limited semantic information and skewed relation distributions. Wang et al. 1 propose VL-SAT, leveraging transformer-based techniques for capturing dependencies and contextual details, enhancing accuracy. Integration of visual and linguistic cues improves performance. The fusion of 2D visual images with 3D point cloud features enables a comprehensive analysis, enhancing semantic-based multi-modal prediction via a frozen CLIP adapter. Recent advancements in large-scale cross-modal pretraining, like CLIP, align 2D image semantics with linguistic semantics, effectively addressing long-tailed issues in tasks related to 2D scene graphs and human-object interaction. However, adapting such assistance to the 3D scenario remains ambiguous. The authors claim that: Proposal of VL-SAT: Introducing a novel Visual-Linguistic Semantics Assisted Training scheme to

enhance discrimination of long-tailed and ambiguous semantic relation triplets in point cloud-based 3DSSG prediction model.

Integration of Multi-Modal Prediction Model: Simultaneously training a powerful multi-modal prediction model (oracle model) aligned with the 3D model, capturing reliable structural semantics through vision data, language signals, and geometric features, and efficiently embedding these benefits into the 3D model through back-propagated gradient flows.

## 2 Scope of Reproducibility

Wang et al. 1, focuses on computing the following metrics (check Appendix B for detailed explanation on the metrics used) on MMGNet (VL-SAT) model to validate their claims:

1. **Evaluation of Object and Predicate Prediction:** The experimental methodology adheres to the protocol outlined in 3DSSG [36], ensuring consistency by placing 3D scenes within the same coordinate system for both training and testing phases.

   - Object and predicate prediction are evaluated using the top-k accuracy ($A@k$) metric. This metric gauges prediction accuracy by considering the top-k predictions generated by the model. Specifically, the scores for subjects, predicates, and objects are multiplied to derive triplet scores, upon which the top-k accuracy is computed. A triplet is deemed correct only if all three components (subject, predicate, and object) are accurately predicted.

2. **Evaluation of Scene Graph and Predicate Classification:** Two 2D scene graph tasks are adapted for the 3D context following the methodology proposed by Zhang et al. 12. These tasks encompass Scene Graph Classification (SGCls) and Predicate Classification (PredCls). SGCls evaluates triplets holistically, while PredCls focuses solely on predicate correctness against ground-truth object labels.

   - Recall at the top-k ($R@k$) triplets serves as the evaluation metric for these tasks, wherein a triplet is considered correct if all components (subject, predicate, and object) are valid.
   - Furthermore, mean recall ($mR@k$) is employed to gauge performance across unevenly sampled relations, akin to $mA@k$. This comprehensive assessment ensures the model's competence in handling long-tailed predicate distributions.

To evaluate the claims made regarding the performance of the VL-SAT method and its sensitivity to critical design choices and hyperparameters, we conducted experiments using a subset of the dataset and various models as outlined in Wang et al. 1. The experiments were carried out with hyperparameters specified by the authors, employing k-fold cross-validation over 100 epochs with validation conducted every 10 epochs.

Our findings indicate that the results obtained for all evaluated metrics are consistently 5-10% lower in performance compared to the reported results in the paper. While we couldn't replicate the exact results reported by the authors across all experiments, it is noteworthy that VL-SAT demonstrates robustness and generally achieves the goals outlined in the paper.

In addition to reproducing the previous work, we thoroughly assessed critical design choices and claims, focusing on two key questions:

1. **Relevance of Proposed Components:** We investigated whether all proposed components, such as the utilization of 2D visual features and the Clip adapter of the VL-SAT architecture, significantly contribute to the performance of the MMGNet model.

2. **Sensitivity to Hyperparameters:** We examined the sensitivity of VL-SAT to critical hyperparameters such as learning rate (lr), weight decay, use of AMSGrad optimizer, and the dimensionality of the clip features (clip_feat_dim). This analysis sheds light on how variations in hyperparameters impact the performance and stability of the VL-SAT method.

Our assessment provides insights into the effectiveness of VL-SAT, the relevance of its design choices, and the impact of hyperparameter tuning on its performance, contributing to a better understanding of its capabilities and limitations.

## 3 Experimental Setup and Code

### 3.1 Generating 3D Indoor Scene Datasets

As per the authors, we are using the 3DSSG indoor scene datasets for scene graph generation. Setting up the environment for generating datasets was provided and was pretty straightforward as we used Ubuntu, a Linux based OS for setting up the conda environment. After setting up the environment, we need to generate some code-specific files such as `relationships.json`, `objects.json`, `train_scans.txt`, and so on, which will be later used for object detection and for classifying the object-predicate-object triplets/relationships. The files generated were stored in the main folder. Then, we requested for 3RScan dataset (for identifying the 3D scans) access which we got pretty soon. We got a `download.py` file which helps to download all the various different types of files needed for setting the 3D scans dataset.

Instead of preparing 1553 data scan folders, we downloaded and prepared 150 data scan folders due to less compute resources. We added in our own bash script for downloading and preparing the data scan folders. The script provided by the authors was erroneous and it took us a really long time to figure this out as initially we thought, we couldn't download and prepare because of low speed internet issues. After downloading the corresponding files, we added code for unzipping the `sequence.zip` folders and added in our bash script for generating aligned instances point cloud information. The file `CVPR2023-VLSAT-reproducibility/data_processing/transform_ply.py` file is used for generating the corresponding aligned instances for all the 150 scan folders. This is required to align the position, color and sets of point cloud data that helps in determining the optimal transformation that maximizes the overlap (IoU; intersection over union)

```
# Generate aligned instance ply
logger_py.info('generate aligned instance ply')
cmd = [
    CVPR2023-VLSAT-reproducibility/data_processing/transform_ply.py,
    "-c", args.config,
    "--thread", str(args.thread)
]
```

Since we are using 150 scan folders, we needed to redefine the `rescans.txt` (file containing all the scan-ids), `train_scans.txt`, `validation_scans.txt`, `relationships_train.json`, and `relationships_validation.json`. We used the files `relationships.json` and `objects.json`. to filter out the 132 scan folders for training and 18 scan folders for validation (the authors also used the same ratio of training:validation). We used our own code (provided in GitHub) to redefine the mentioned files.

We redefined the training data source location, output location and few other file locations in the `magnet.json` (the configuration file used for training the model MMGNet) to start the training. We used the MMGNet model for training that was provided by the authors. As per our request, the authors added SGFN, SGPN and SGGPoint models in the original GitHub code.

### 3.2 Model(s) Architecture

VL-SAT along with SGFN, SGGPoint models uses the PointNet 3D feature detection algorithm MMGNet uses PointNet (Appendix A) architecture for 3D object classification and semantic segmentation. The PointNet architecture is modified by the authors in the pointnet network algorithm that uses a Graph convolutional network for object(s) detection, classification and segmentation. We faced some issues while training the model MMGNet as the GraphTripleConvNet wasn't available under the src/lib directory. We found the particular graph convolutional network in Google's sg2im (Generating Scene Graphs from 2D images) and

used the same graph convolutional neural network defined in their `graph.py` file in our code for implementing the PointNet architecture.

Figure 1: Overview of the Visual-Linguistic Semantics Assisted Training (VL-SAT) framework for 3D scene graph prediction. Adapted from Wang et al. 1 VL-SAT incorporates 2D visual data and linguistic semantics during training to enhance 3D scene graph predictions through node and edge-level collaboration and triplet-level regularization. During inference, VL-SAT utilizes only 3D point cloud data to generate reliable 3D scene graphs.

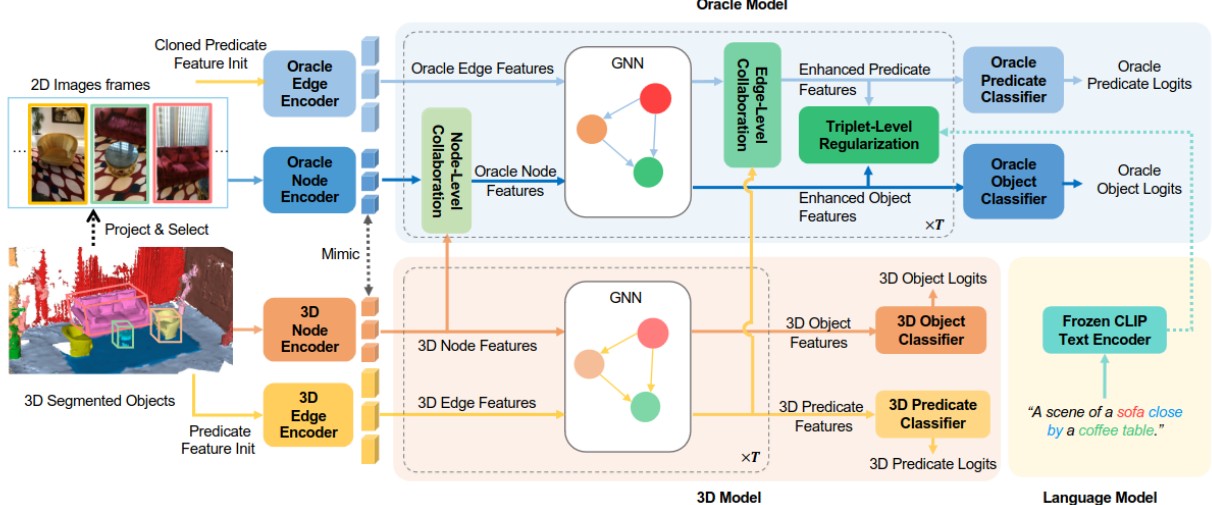

Official Pytorch Geometric repository made some changes in defining the function names used for convolutional neural networks for creating message passing layers. Message Passing algorithm in Graph neural network enables information exchange and aggregation among nodes in a graph. Since the authors used the same message passing algorithm (as a parent class) used in pytorch geometric module for generating edge descriptor function (in `src/model/model_utils/network_util.py`), generating edge indices' function (in `src/utils/op_utils.py`) and aggregation of the edge indices' functions (in `src/utils/op_utils.py`) (child class), we renamed the functions such as _check_input(), _set_size(), _index_select() and _collect().

## 3.3 Training Various Models

The baseline models for each of the models are present in `src/models/`. The authors provided us the code for the baseline model for MMGNet, the derived model which derives the class from the baseline model and introduces the training, validation and evaluation metrics and the main file that loads the MMGNet model provided with the bash command from terminal. The baseline code for other models was provided by the authors. We created derived models for SGFN, SGGPoint from the baseline models taking inspiration from the MMGNet model. We created varying `main.py` files to run those different models hence creating a pipeline for evaluating the models and reproducing the results and checkpoints in the paper.

We created varying configuration files for each of the models which are in the `config/` folder.

The results are stored in results under varying model names, checkpoints in `ckp` folder, logs and configurations respectively in their folders. We provide all the model configurations, results, checkpoints as additional materials.

## 3.4 Ablation Studies

To assess the contribution of different components to the VL-SAT architecture, we conducted experiments focusing on various aspects while comparing them to the baseline VL-SAT model. Here are the variations we considered:

- Node-level Collaboration: We examined the role of node-level collaboration in VL-SAT by omitting interactions between nodes from both 3D and 2D node encoders. This analysis highlighted the impact of collaboration on model performance, with extracted features termed as oracle node features for subsequent processing by the oracle GNN.

- Edge-level Collaboration: After extracting 2D edge features from the oracle GNN and 3D predicate features from the 3D model, collaboration occurs at the edge level. Conversely, we investigate the model's performance when the mechanism enabling interactions between edges, which represent relationships or connections in the graph, is removed.

- Triplet-level CLIP-based Regularization: An ablation study was conducted to assess the impact of triplet-level CLIP-based regularization within the VL-SAT architecture. Triplet-level CLIP-based Regularization process involves integrating enhanced predicate and object features from the oracle model with a frozen CLIP text encoder, culminating in the creation of an oracle-level object and predicate classifier.

- Cross-modal Attention: The ablation study on cross-modal attention investigates the simultaneous use of node collaboration and edge collaboration within the VL-SAT architecture.

- Multi-Head Self-Attention (MHSA): In the ablation study on Multi-Head Self-Attention (MHSA), we analyze the impact of incorporating MHSA into both the graph edge attention network and the graph node attention network within the VL-SAT architecture.

By examining these variations and comparing their performance to the baseline VL-SAT model, we gained insights into the specific contributions of the Clip adapter and 2D image features in addressing the objectives outlined in our reproducibility scope.

## 3.5 Hyperparameter Tuning

In addition to conducting all the ablation studies, we also performed hyperparameter tuning to optimize the model's performance. In all experiments and ablations, we adhere to the hyperparameters specified in the original paper. However, we also conduct hyperparameter search to optimize certain parameters such as learning rate from 0.0001 to 0.0003, AMSGrad, and AdamW weight decay, aiming for improved accuracy. We experiment with various values for these parameters and record the best-performing values in the table for reference.

Dropout is a regularization technique used to prevent overfitting by randomly dropping out neurons during training. We have a dropout rate of 0.5 (50%), then during each training iteration, each unit in the dropout layer has a 50% chance of being temporarily "dropped out" or set to zero. This means that the network cannot rely too heavily on any individual neuron and must learn redundant representations, thus improving its generalization ability.

AMSGrad is an extension of the Adaptive Moment Estimation (Adam) optimizer that aims to address the issue of adaptive learning rates growing too large in some dimensions. Since the performance with AMSgrad is comparable to the other optimizers, it is recommended to continue using it unless there is a significant improvement in performance by switching to another optimizer.

Weight decay (w.decay = false), also known as L2 regularization, is used to prevent overfitting by adding a penalty term to the loss function. In the table, weight decay is turned off (w.decay = false). You may want to try turning it on and experimenting with different decay rates to observe its effect on the model's performance.

In the context of CLIP, these feature vectors represent both images and text in a shared embedding space, allowing the model to understand the relationships between visual and textual information. By increasing the dimensionality of these feature vectors from 512 to 1024, we are effectively increasing the complexity and richness of the representations learned by the CLIP model.

In the multi-head attention mechanism, the number of heads refers to the number of parallel attention layers that operate independently and in parallel. Each attention head focuses on different parts of the input data and produces its own output. Increasing the number of heads allows the model to capture more diverse relationships and dependencies in the data. Typically, the number of heads is a hyperparameter that can be adjusted during model training.

In some training procedures, it's common to freeze certain layers or components of a model to prevent them from being updated during training. This can help stabilize the training process, prevent overfitting, or allow specific parts of the model to converge more effectively. By setting "update_2d" to false, it's indicated that the 2D features, which could represent visual features extracted from images, are not being updated during the current epoch.

### 3.6    Computation Resources

In this study, we used a single NVIDIA GeForce GTX 3090 GPU for conducting all experiments, including training different models, hyperparameter tuning, and performing ablation studies. The total runtime required for these tasks amounted to approximately 6 GPU hours per experiment, totaling 96 GPU hours across all experiments. Due to the limitations of using a single compute resource, parallel processing was not feasible, necessitating the execution of one model at a time. Although we received compute resources from ML Collective approximately 4-5 days before submission, we were unable to fully use them within the given timeframe.

## 4    Results

In this section, we present the results of our experimental runs using VL-SAT and other pertinent models, along with their ablations and hyperparameter studies. We have structured our Figure 2 following the format employed by Wang et al. 1, ensuring easy comparison among the different models. However, we created the other two tables independently. Figure 1 encompass a range of models, including VL-SAT, SGFN, VLSAT-oracle, and non-VLSAT, facilitating comprehensive analysis. Additionally, we have incorporated additional ablation studies such as node collaboration, edge collaboration, and triplet-level clip regularization in Figure 3. These studies aim to ascertain the impact of various factors on model performance, such as determining the optimal graph convolutional network and evaluating the contribution of the multi-head attention network to the VL-SAT model. Regarding hyperparameters, we explored various values for parameters such as learning rate, weight decay, and dropout of the attention network, aiming to optimize the predictive performance of the VL-SAT model in Figure 4.

Figure 2: We present the performance metrics of VL-SAT alongside SGFN, VLSAT-oracle, and VLSAT (only 3d). By aligning our results with the structure established by Wang et al. 1. We aim to provide a clear overview of each model's efficacy. Notably, we highlight any deviations between our findings and those reported by the authors, elucidating their implications for the respective claims made.

| Model | Object | | | Predicate | | | | | | Triplet | | | |
|---|---|---|---|---|---|---|---|---|---|---|---|---|---|
| | A@1 | A@5 | A@10 | A@1 | A@3 | A@5 | mA@1 | mA@3 | mA@5 | A@50 | A@100 | mA@50 | mA@100 |
| non-VL-SAT | 44.13 | 67.16 | 75.63 | 82.04 | 94.96 | 98.4 | 25.62 | 54.47 | 70.08 | 83.29 | 86.5 | --- | --- |
| VL-SAT | 44.73 | 68.05 | 75.33 | 82.19 | 95.05 | 98.52 | 24.01 | 52.79 | 70.36 | 83.58 | 86.87 | --- | --- |
| VL-SAT (oracle) | 45.02 | 67.16 | 76.23 | 82.33 | 95.52 | 98.45 | 27.2 | 56.23 | 70.94 | 86.89 | 89.98 | --- | --- |
| SGFN | 45.02 | 66.12 | 75.78 | 84.16 | 95.85 | 98.42 | 28 | 56.61 | 69.67 | 82.97 | 85.42 | --- | --- |
| SGGPoint | 45.17 | 68.35 | 77.41 | 84.5 | 94.28 | 97.26 | 18 | 35.83 | 46.59 | 82.96 | 86.29 | --- | --- |

### 4.1 Results reproducing original paper

VL-SAT: VL-SAT achieves the best performance in several metrics, particularly in 3D Triplet Accuracy@50 and 3D Triplet Accuracy@100, where it outperforms the other models by approximately 7-8%. It also performs best in 3D Relation Accuracy@5 and 3D Mean Relation Acc@5, surpassing other models by around 1-2%.

SGFN: SGFN performs relatively worse compared to VL-SAT in most metrics. It lags behind by around 30% in 3D Triplet Accuracy@50 and 3D Triplet Accuracy@100. Additionally, it falls behind by approximately 13% in 3D Mean Relation Acc@1.

SGGPoint: SGGPoint performs competitively in most metrics but falls short compared to VL-SAT. It trails behind by around 25% in 3D Triplet Accuracy@50 and 3D Triplet Accuracy@100. Additionally, it lags by approximately 5% in 3D Mean Relation Acc@5 compared to VL-SAT.

VL-SAT-Oracle: VL-SAT-Oracle performs similarly to VL-SAT, as it is designed to provide an upper bound for VL-SAT's performance. The difference between VL-SAT and VL-SAT-Oracle is negligible in most metrics.

Non-VLSAT: This model's performance is comparable to VL-SAT in most metrics, with only slight variations. It performs nearly on par with VL-SAT in all metrics, with minimal differences.

VL-SAT emerges as the best-performing model across various metrics, showcasing its effectiveness in 3D scene graph prediction tasks. Regarding VLSAT, our results of the object, predicate, triplet scores are 5-10% diverging for each of the models. Although, we used a subset of dataset, the metrics are kfold cross validated hence, our models don't seem to be undercutting. Yet, those numbers reported by the authors are varying than ours but at the same time supporting their first (1) claim that VL-SAT performs better than any other model.

### 4.2 Results of ablation studies - beyond the paper

In this section, we conduct a detailed performance analysis and ablation study of different components of the VL-SAT model. We compare the performance of VL-SAT with variations such as NC(node-level collaboration), EC(edge-level collaboration), TR(triplet-level CLIP-based regularization), VL-SAT without Multi-Head Self-Attention (MHSA), and TRIP Graph Convolutional Network (GCN TRIP). By comparing these components of the model, we aim to understand the contribution of each part to the overall performance of the VL-SAT model in accurately predicting 3D scenes.

Figure 3: We outline the outcomes of our ablation study, where we systematically evaluate the impact of different components within VL-SAT and related models. Through meticulous analysis, we dissect the contributions of individual elements, shedding light on their significance in model performance. Any disparities observed in comparison to the original study are emphasized, underscoring potential variations in experimental setups or model configurations.

| Model | Object | | | Predicate | | | | | | Triplet | | | |
|---|---|---|---|---|---|---|---|---|---|---|---|---|---|
| | A@1 | A@5 | A@10 | A@1 | A@3 | A@5 | mA@1 | mA@3 | mA@5 | A@50 | A@100 | mA@50 | mA@100 |
| VL-SAT (TR) | **45.02** | 67.16 | **76.23** | 82.33 | **95.52** | **98.45** | 27.2 | 56.23 | **70.94** | 14.89 | 19.98 | --- | --- |
| VL-SAT(MHSA) | 40.71 | 64.78 | 75.48 | 81.79 | 95.12 | 98.02 | 19.68 | 41.01 | 51.59 | 81.46 | 84.4 | --- | --- |
| EC | 42.79 | **67.45** | 75.93 | **82.77** | **95.88** | **98.44** | **27.89** | **58.18** | 69.73 | 82.54 | 86.08 | --- | --- |
| NC | 42.13 | 65.16 | 73.63 | 82.04 | 94.96 | 98.4 | 25.62 | 54.47 | 70.08 | 83.29 | 86.5 | --- | --- |
| GCN (TRIP) | **44.87** | 67.31 | **76.37** | **82.44** | 95.5 | **98.49** | **27.25** | **56.63** | 70.71 | **83.63** | **87.07** | --- | --- |
| VL-SAT | **44.13** | 67.16 | **75.63** | 82.04 | 94.96 | 98.4 | 25.62 | 54.47 | 70.08 | 83.29 | 86.5 | --- | --- |

### 4.2.1 Ablation Analysis Summary

For VL-SAT (TR), there's an improvement of approximately 0.89% in Object A@1, 0.60% in Object A@10, 0.60% in Predicate A@1, 0.29% in Predicate A@3, 0.49% in Predicate A@5, and 2.45% in Triplet mA@1. However, there's a slight decrease of 0.24% in Object A@50 and 0.22% in Object A@100.

VL-SAT (MHSA) shows a degradation of approximately 3.42% in Object A@1, 2.38% in Object A@5, 0.15% in Object A@10, 0.24% in Predicate A@1, 0.81% in Predicate A@5, and 2.08% in Triplet mA@1.

EC demonstrates an improvement of approximately 1.66% in Object A@1, 0.30% in Object A@10, 0.73% in Predicate A@1, 0.29% in Predicate A@3, 0.41% in Triplet mA@1, and 0.12% in Triplet mA@3. However, there's a slight decrease of 0.75% in Object A@50 and 0.42% in Object A@100.

NC performs similarly to the baseline in most metrics, with slight improvements of approximately 0.13% in Predicate A@1, 0.38% in Predicate A@3, and 0.12% in Triplet mA@1.

GCN (TRIP) exhibits improvements of approximately 0.74% in Object A@1, 0.74% in Object A@10, 0.24% in Predicate A@1, 0.42% in Predicate A@3, 0.39% in Predicate A@5, and 2.57% in Triplet mA@1. There are minor improvements of approximately 0.40% in Object A@50 and 0.57% in Object A@100.

We observe that VL-SAT (TR) and GCN (TRIP) achieve the highest performance in terms of A@1, A@5, and A@10 metrics, indicating that the triplet-level CLIP-based regularization and TRIP Graph Convolutional Network rather than EAN GCN features contribute significantly to the model's accuracy. VL-SAT without Multi-Head Self-Attention (MHSA) shows lower performance across all metrics, suggesting that this component is extremely important to have more number of heads for graph scene prediction. The notable improvements observed across all metrics underscore the significant impact of both edge and node collaboration in enhancing the performance of the VL-SAT architecture model. This suggests that leveraging collaborative learning strategies, such as edge and node collaboration, can substantially contribute to the overall effectiveness and efficiency of the model. By fostering cooperation and interaction among interconnected entities, these collaborative approaches enable the VL-SAT model to capitalize on the collective intelligence and insights derived from collaborative efforts, ultimately leading to improved performance across various evaluation metrics. Further analysis is needed to understand the impact of each component on the overall performance of the VL-SAT model. Despite some variations between the reported numbers by the authors and our own findings, the results still align with the authors' secondary assertion that multi-modal learning in VL-SAT's architecture outperforms other models.

### 4.3 Results regarding hyperparameter analysis - beyond the paper

In this section, we explore the impact of various hyperparameters on fine-tuning the baseline VL-SAT model. Specifically, we investigate adjustments to the learning rate, using AMSGrad, toggling weight decay, and introducing dropout attention. Through systematic experimentation and analysis, we aim to discern how these modifications influence the performance of the VL-SAT model, providing insights into optimal configurations for enhancing its effectiveness in 3D semantic scene graph prediction.

In Figure 4, the baseline VL-SAT model achieved an mAP@1 of 44.13% and mA@100 of 98.4%. Configurations with higher learning rates (lr=0.0003) generally led to improved performance, with the highest mAP@1 and mA@100 achieved at 49.33% and 98.85%, respectively. Optimizing with AMSGrad and disabling weight decay maintained competitive results compared to the baseline, with a slight improvement in mAP@1 and a marginal decrease in mA@100. However, adding dropout to attention (dropout attn=0.25) resulted in decreased performance across all metrics compared to the baseline VL-SAT model.

Using a learning rate of 0.0003 seems to bring the results closer to those reported in the original paper. This adjustment in the learning rate might help the model converge more effectively during training, leading to performance metrics that align better with the expected outcomes described in the paper. It's essential to fine-tune hyperparameters like learning rate to achieve optimal performance and match the expected results. We performed an extensive hyperparameter tuning to check that weight decay with amsgrad decreases the performance but with a learnign rate of 0.0005, some of the metrics improve by a little percentage hence, concluding that 0.0003 is the ideal learning rate for VL-SAT.

Figure 4: We present the results of our hyperparameter study, elucidating the effects of varying key parameters on model performance. Through rigorous experimentation, we explore the sensitivity of VL-SAT and its counterparts to different hyperparameter configurations. We emphasize any differences in outcomes compared to the reference study, offering insights into the robustness of the models under different settings.

| Model | Object | | | Predicate | | | | | | Triplet | | | |
|---|---|---|---|---|---|---|---|---|---|---|---|---|---|
| | A@1 | A@5 | A@10 | A@1 | A@3 | A@5 | mA@1 | mA@3 | mA@5 | A@50 | A@100 | mA@50 | mA@100 |
| lr-0.0003 | **49.33** | 70.58 | 79.79 | **85.55** | **97.1** | **98.85** | **44.69** | **68.16** | **78.6** | **84.82** | **87.77** | --- | --- |
| lr-0.0003, amsgrad=true, w.decay = false | 48.74 | **71.77** | **80.68** | 85.18 | 96.42 | 98.68 | 41.05 | 63.64 | 76.55 | 84.45 | 87.77 | --- | --- |
| dropout attn=0.25 | 45.62 | 67.61 | 77.86 | 83.3 | 95.47 | 98.45 | 26.05 | 57.72 | 71.39 | 84.24 | 87.54 | --- | --- |
| VL-SAT | 44.13 | 67.16 | 75.63 | 82.04 | 94.96 | 98.4 | 25.62 | 54.47 | 70.08 | 83.29 | 86.5 | --- | --- |

# 5 Discussion

While the study is largely replicable due to the availability of the authors' code, it falls short of being readily reproducible. Despite the code being open-source, it contains several errors, lacks sufficient documentation, and does not offer a straightforward method to reproduce the reported results. Merely using the provided code is insufficient, and attempts to extend it according to the paper's specifications often result in divergent numerical outcomes across multiple experiments. Consequently, our assessment concludes that the paper lacks reproducibility in terms of obtaining the reported numbers as we got nearly a 5-10% decrease in percentage of each of the metrics as in the original paper.

However, it's important to note that the majority of our experiments which were ablation studies support the primary claims made in the paper. Despite conducting hyperparameter tuning, we observed minimal impact on the model's performance, with no significant improvements noted.

## 5.1 Limitations

Our attempt to replicate the study encountered several hurdles stemming from disparities in the dataset generation script and inconsistencies in the training procedures. To address these issues, we made necessary adjustments, including replacing PointNet graph network with SG2IM graph network. However, these modifications led to deviations from the original experimental setup. Additionally, our replication efforts were hampered by constraints on computational resources, limiting our ability to explore further experiments.

## 5.2 What was easy

It was straightforward for us to grasp the main claims of the paper and the experiments supporting them. Setting up the development environment and generating datasets were relatively easy tasks. This was mainly due to the authors' provision of dataset links and code for data loaders, model definitions, and training procedures. With all the necessary code provided, training models and calculating metrics like top-k accuracy, recall, and zero-shot recall became seamless processes.

## 5.3 What was difficult

Setting up the environment for generating datasets and preparing the 3RScan dataset was relatively straightforward, thanks to the provided instructions and scripts. However, we encountered challenges when discrepancies arose in the script provided for downloading and preparing data scan folders. This led to delays as we had to troubleshoot and resolve the issues. Additionally, due to our limited computational resources, we had to reduce the number of data scan folders from 1553 to 150. During model training, we faced further

complexities, especially with the MMGNet architecture. The absence of GraphTripleConvNet in the designated directory was a significant hurdle. To overcome this, we had to utilize a similar graph convolutional network from Google's sg2im repository. Furthermore, changes in function names within the PyTorch Geometric repository required us to rename several functions to ensure compatibility with the message passing algorithm used in our code.

### 5.4 Communication

Throughout the reproducibility project, we maintained communication with the original authors. They provided us with the baseline code for sgpn, sgfn, and sggpoint models, which we utilized for comparison purposes. Despite our repeated inquiries about various issues and discrepancies observed among the models, we did not receive a response. We are grateful for their assistance in replicating their paper; however, we believe that documenting clarifications within the code or paper, as well as sharing the generated datasets through file-sharing mechanisms, could have enhanced the reproducibility process.

### 5.5 Acknowledgments

We extend our gratitude to the ML Collective team for providing valuable compute resources during our project. Additionally, we appreciate Ziqin Wang for generously sharing the code for various other models, which proved to be instrumental in our research efforts.

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

# A  PointNet library

PointNet, a novel neural network designed to directly process point cloud data, preserving the inherent permutation invariance. Unlike traditional methods that convert point clouds into voluminous voxel grids or image collections, PointNet offers a unified architecture for various applications, including object classification, part segmentation, and scene semantic parsing.

# B  The evaluation metrics

The metrics evaluated in this context include:

1. Top-k Accuracy (A@k): - This metric evaluates the accuracy of object, predicate, and triplet predictions. - It considers the top-k predictions generated by the model, where 'k' denotes the number of predictions taken into account. - A higher A@k score signifies greater accuracy, indicating that more of the model's top predictions align with the ground truth.

2. Mean Top-k Accuracy (mA@k): - This metric calculates the average top-k accuracy across all predicate classes. - It offers an assessment of the model's performance across various predicate categories, capturing the variability in the distribution of predicate classes.

3. Recall at the Top-k (R@k): - R@k evaluates the correctness of triplets by examining the top-k predictions produced by the model. - A triplet is deemed correct if all three elements (subject, predicate, and object) match the ground truth. - This metric measures the model's efficacy in retrieving relevant information from the predicted triplets.

4. Mean Recall (mR@k): - Similar to mA@k, mR@k focuses on assessing the model's performance in capturing unevenly sampled relations. - It provides a holistic evaluation of the model's capability to handle long-tailed predicate distributions, offering insights into its robustness across varied relation types.

These metrics collectively provide a comprehensive evaluation of the model's performance in object and predicate prediction, scene graph classification, and predicate classification tasks within the 3D scene graph scenario.

