# OpenReview forum: "[RE] A Reproducibility Study on Scene-Graph Generation from 3D Point Clouds: Hybrid Approach with Clip, 2D Image Semantics, and 3D Geometry"
_TMLR — Rejected by TMLR_

### Review · Reviewer_b5Xh · 2024-05-06

**Summary Of Contributions:**

The paper aims to reproduce and validate the findings of a previous study on 3D semantic scene graph prediction using a hybrid approach. The authors leverage visual and linguistic semantics to replicate the top-k accuracy results and explore models such as SGFN and SGGPoint. They also assess the impact of a CLIP adapter, 2D image semantics, and conduct hyperparameter tuning. The study includes an ablation study to investigate node and edge collaboration and the influence of a multi-head self-attention network within the VL-SAT architecture.

**Audience:**

Yes

**Broader Impact Concerns:**

There is no Broader Impact Concerns.

**Claims And Evidence:**

Yes

**Requested Changes:**

1. Detailed Experimental Setup: The paper should include a more detailed description of the experimental setup to address the discrepancies in the results.
2. Extended Computational Resources: If possible, the authors should seek additional computational resources to handle the full dataset to ensure the results are not biased by reduced data.
3. Improved Communication: Encourage ongoing communication between the reproducing authors and the original authors to clarify any issues or discrepancies.
4. Hyperparameter Analysis: Conduct a more extensive hyperparameter sensitivity analysis to better understand the model's behavior and potentially identify more optimal settings.
5. Comparison with State-of-the-Art: The paper could benefit from a comparison with other state-of-the-art methods to contextualize the performance of the VL-SAT model.
6. Error Analysis: Include an error analysis section to understand the types of mistakes the model is making and potential ways to address them.
7. Long-Term Training Curves: Provide long-term training curves and model checkpoints to help future researchers understand the training dynamics and for further validation of the model's performance.

**Strengths And Weaknesses:**

Advantages:

1. Comprehensive Methodology: The paper uses a robust methodology that includes reproducing results, conducting ablation studies, and hyperparameter tuning.
2. Open-Source Collaboration: The authors collaborate with the original authors and utilize open-source code, which enhances the reproducibility and transparency of the study.
3. In-Depth Analysis: The paper provides a detailed analysis of the VL-SAT architecture, including the impact of different components and hyperparameters on the model's performance.
4. Replicability: Despite challenges, the study is largely replicable, which is a testament to the authors' diligence in recreating the experimental setup.

Disadvantages:

1. Discrepancies in Results: There are notable discrepancies between the values obtained by the authors and those reported in the original paper, indicating a lack of precise reproducibility.
2. Communication Gaps: The lack of response from the original authors after the initial code integration hindered the resolution of discrepancies.
3. Hyperparameter Sensitivity: The study found minimal impact from hyperparameter tuning, suggesting potential limitations in the model's sensitivity to these parameters.

---

### Review · Reviewer_neVE · 2024-05-07

**Summary Of Contributions:**

The paper presents a reproducibility study on the VL-SAT method, encompassing model reproduction, extensive ablation studies, and hyperparameter tuning. It validates the correctness and effectiveness of the VL-SAT approach.

**Audience:**

Yes

**Broader Impact Concerns:**

There are no additional concerns regarding the ethical implications of the work.

**Claims And Evidence:**

Yes

**Requested Changes:**

The authors are advised to redo the experiments on the entire 1553 scenes of the 3RScan/3DSSG dataset to ensure more reliable and comparable results.

**Strengths And Weaknesses:**

Strengths:

The paper provides a careful re-implementation of VL-SAT.
It conducts a thorough examination of ablations and hyperparameter tuning.

Weaknesses:

Training Data Discrepancy: The original VL-SAT utilized all 1553 scenes in the 3RScan dataset, whereas this study only employs 150 scenes, potentially leading to inconsistencies in the results.

Computing Resources: The claim of insufficient computing resources is questionable. The original VL-SAT experiments were conducted on a 2080 Ti GPU over 48 hours for the full dataset. Given that the reproducibility study has access to a 3090 GPU, the assertion of lacking resources does not hold.

Inappropriate Figure Representation: Figures 2 and 3 are presented as snapshots rather than being formatted as tables. For clarity and consistency, these should be converted into LaTeX tables.

---

### Review · Reviewer_rEQG · 2024-08-11

**Summary Of Contributions:**

This paper examines the reproducibility of VL-SAT, a method for 3D scene graph prediction presented at CVPR 2023. While the authors provide a detailed evaluation of this specific method, the study's narrow focus limits its broader impact. The main findings resemble a technical report, concluding that VL-SAT can be reproduced with additional code modifications.

**Audience:**

No

**Broader Impact Concerns:**

NA.

**Claims And Evidence:**

Yes

**Requested Changes:**

To enhance its relevance, the paper should expand its scope to include multiple methods within 3D scene graph prediction. Evaluating various approaches would allow the authors to offer comparative insights into the challenges and best practices for reproducibility in this field. This could involve analyzing different data preprocessing techniques, architectural variations, and evaluation metrics to better understand the factors influencing reproducibility.

Moreover, exploring broader lessons from the reproducibility process could enrich the study. Identifying common challenges and innovative solutions across different methods would provide valuable insights into best practices. These aspects would appeal to a wider audience interested in improving reproducibility in machine learning and computer vision.

**Strengths And Weaknesses:**

Pros:
1. Comprehensive evaluation of VL-SAT.

Cons:
1. Focuses on only one method.
2. Provides limited added value and insights to the community.

---

### Decision · Action_Editor_LvCK · 2024-10-11

**Recommendation:** Reject

**Comment:**

The reviewers provided detailed comments. However, the author only responded to the comments but did not attempt to revise the manuscript.  Thus reviewers recommended reject.

**Audience:**

The manuscript provides limited added value in terms of broader lessons, which would be more appealing to a wider audience.  E.g., error analysis, comparisons with other SOTA methods, hyperparameter analysis, etc.

**Claims And Evidence:**

The reproducibility claim is not convincing since less data was used for training in this reproducibility study, and thus leading to inconsistent results with the original paper.